# Therapeutic Potential of Alpha-1 Antitrypsin in Type 1 and Type 2 Diabetes Mellitus

**DOI:** 10.3390/medicina57040397

**Published:** 2021-04-20

**Authors:** Sangmi S. Park, Romy Rodriguez Ortega, Christina W. Agudelo, Jessica Perez Perez, Brais Perez Gandara, Itsaso Garcia-Arcos, Cormac McCarthy, Patrick Geraghty

**Affiliations:** 1Department of Medicine, State University of New York Downstate Health Sciences University, Brooklyn, NY 11203, USA; Sangmi.Park01@downstate.edu (S.S.P.); Romy.RodriguezOrtega@downstate.edu (R.R.O.); Christina.Agudelo@downstate.edu (C.W.A.); Jessica.PerezPerez@downstate.edu (J.P.P.); Brais.PerezGandara@downstate.edu (B.P.G.); Itsaso.Garcia-Arcos@downstate.edu (I.G.-A.); 2University College Dublin School of Medicine, Education and Research Centre, St. Vincent’s University Hospital, D04 T6F4 Dublin, Ireland; cormac.mccarthy@ucd.ie

**Keywords:** alpha-1 antitrypsin, diabetes mellitus, inflammation, apoptosis

## Abstract

Alpha-1 antitrypsin (AAT) has established anti-inflammatory and immunomodulatory effects in chronic obstructive pulmonary disease but there is increasing evidence of its role in other inflammatory and immune-mediated conditions, like diabetes mellitus (DM). AAT activity is altered in both developing and established type 1 diabetes mellitus (T1DM) as well in established type 2 DM (T2DM). Augmentation therapy with AAT appears to favorably impact T1DM development in mice models and to affect β-cell function and inflammation in humans with T1DM. The role of AAT in T2DM is less clear, but AAT activity appears to be reduced in T2DM. This article reviews these associations and emerging therapeutic strategies using AAT to treat DM.

## 1. Introduction

Alpha-1 antitrypsin (AAT) has established anti-inflammatory and immunomodulatory effects that go beyond its anti-protease activity, which are well documented in the pathogenesis of emphysema [1]. Although a large portion of the research in AAT has focused on pulmonary and liver diseases, there is a growing body of evidence linking AAT with other disease processes with an immune/inflammatory component [2,3,4,5]. Immune dysregulation and inflammation play a significant role in type 1 diabetes mellitus (T1DM) and is a subject of intense study for more than 100 years when Schmidt in 1902 described a peri-islet cellular infiltrate in the pancreas of a deceased 10-year-old child with diabetes [6,7,8,9]. Targeting immune/inflammatory pathways is a common and active area of research in the treatment of T1DM, which includes the use of agents such as glucocorticoids, cyclosporine, azathioprine, anti-thymocyte globulin, and rituximab but also cell therapies with varying results [10,11,12,13,14,15]. AAT’s association with diabetes mellitus started to become apparent in the 1980s when AAT activity was found to be reduced in patients with diabetes [16]. Since then, there is accumulating evidence of AAT’s role in T1DM [17,18,19] and a probable, but less clear role, in type 2 diabetes mellitus (T2DM) [20]. Here we review these associations and the potential for use of AAT targeted therapies as a strategy for the prevention and treatment of type 1 and type 2 diabetes mellitus.

## 2. Alpha-1 Antitrypsin Responses Play a Role in Type 1 Diabetes Mellitus

The AAT protein (encoded by the SERPINA1 gene) is primarily produced by the liver and circulated via the blood. AAT is an acute stress reactant protein and increases during stress conditions [21] and during the late stages of pregnancy [22]. AAT is primarily known as a potent anti-elastase protein [23]. AAT deficiency is extensively researched and many genetic variants of AAT are reported [24,25]. The most frequent and clinically significant alleles of AAT are PI*M, PI*S, and PI*Z, with PI*M being the normal/nonmutated allele [26]. The PI*S and PI*Z mutations account for approximately 95% of AAT deficiency cases. Carrying two copies of the severe PI*Z allele mutation leads to misfolding of the AAT protein, AAT protein retention in the liver, low serum levels of AAT (termed AAT deficiency) [27], and thus an inability to regulate inflammatory and proteolytic responses. Accumulation of the malformed protein in the liver leads to local damage [28]. Other clinical manifestations such as emphysema, panniculitis, and Wegener’s granulomatosis can occur in AAT deficient individuals [29]. Notably, AAT responses are significantly lower in the plasma of T1DM patients [16], and plasma anti-trypsin capability progressively decreases with a longer duration of diabetes [30]. Raising blood levels of AAT with augmentation therapy prevents T1DM development, prolongs islet allograft survival [18], increases insulin release capacity [31], and inhibits pancreatic B-cell apoptosis [32]. AAT treatment also significantly reduces HbA1c levels [33]. Importantly, expression of the human nonmutated AAT gene, by a recombinant adeno-associated virus, in nonobese diabetic (NOD) mice significantly reduced insulitis and prevented the development of overt hyperglycemia [17]. Serum AAT levels are significantly reduced in NOD mice [17] and AAT treatment expands the functional mass of the β cells in diabetic NOD mice [34], possibly due to changes in the inflammation milieu. Therefore, clinical and animal models suggest that AAT plays a major role in T1DM in addition to its established role in emphysema in AAT deficient individuals.

## 3. Alpha-1 Antitrypsin Might Have a Role in Type 2 Diabetes Mellitus Pathogenesis

Equally, another report highlighted an association of AAT deficiency with an increased risk of developing Type 2 diabetes (T2DM) [20]. High levels of degraded AAT are observed in the urine of T2DM patients with diabetic kidney disease [35]. T2DM often occurs with other comorbidities that include obesity and nonalcoholic fatty liver disease (NAFLD, or hepatic steatosis). In a clinical study, hepatic steatosis in the presence of acute pancreatitis resulted in reduced AAT levels that correlated with increased disease severity [36]. Alternatively, anti-trypsin capacity is lower in serum from gestational diabetes mellitus compared to healthy pregnant women [37]. However, additional data is needed on whether AAT has a direct role in T2DM pathogenesis and other types of diabetes.

## 4. What Mechanisms Could AAT Regulate in Preventing Diabetes Onset and Progression?

Despite AAT deficiency being extensively studied and exogenous AAT administered as a therapy to AAT-deficient emphysema subjects, little is known about the potential long-term effects of AAT signaling on T1DM patients. However, we do know several possible mechanisms that AAT signaling could mediate in preventing T1DM. AAT is primarily known to be an inhibitor of neutrophil protease, such as proteinase-1, elastase, thrombin, and trypsin [38]. In recent years, multiple roles of AAT beyond its anti-protease capacity have emerged [1,39], including activation of phosphatases [40], inhibition of caspase activity [41] and nitric oxide production [42], subduing HIV type 1 [43] and rhinovirus infectivity [44], reducing endoplasmic reticulum stress responses [45,46], regulating neutrophil degranulation [47,48], modifying dendritic cell maturation and promoting regulatory T cell (Treg) differentiation [49], increasing IL-10 and IL-1Ra release [50], minimizing epithelial barrier damage, and regulating IL-8-mediated neutrophil chemotaxis [51]. AAT also protects multiple proteins from undergoing proteolytic cleavage, such as phospholipid transporter protein (PLTP) [52]; thereby reducing lung inflammation and neutrophil degranulation [53]. AAT can counter damage associated with hypoxia [54,55,56,57], or in clinical circumstances with an underlying allogeneic background [58,59] or danger-associated molecular pattern agents (DAMPs) induced inflammation [60,61]. Equally, neutrophils isolated from human T1DM and T2DM subjects or mice are primed to produce neutrophil extracellular traps (NETs) [62], which could influence neutrophil responses in the lungs. Neutrophils from AAT deficient subjects have increased neutrophil responses [47,51,53] and AAT could counter elevated neutrophil NETs formation and degranulation. Therefore, the potential mechanisms for AAT’s protective role in T1DM could be vast.

Since dysfunctional inflammation is linked to the pathogenesis of T1DM, research was conducted on the beneficial effects of AAT in T1DM in terms of AAT’s anti-inflammatory properties. Indeed, AAT has anti-inflammatory and immune-modulating properties associated with enhancing pancreatic β-cell function, via regulation of IL-1β responses and the development of antigen-specific T regulatory cells [63]. AAT-treated mice have reduced serum TNF-α, lymphocytic infiltration, NF-κB activation, and JNK phosphorylation in their pancreatic β-cell islets [64]. One possible means of AAT regulating inflammation responses is through its regulation of cyclic adenosine monophosphate (cAMP). Increased insulin release capacity due to AAT stimulation coincides with elevated cAMP [31], as does AAT-induced phosphatase activity [40]. Elevated protein phosphatase activity following AAT stimulation attenuated cytokine and protease responses within the lungs [40]. AAT can also regulate nitric oxide responses, a known inducer of islet β cell death, by reducing its release [18]. Equally, AAT prevents apoptosis of pancreatic B-cells [32] and lung cells [65] by inhibiting caspase-3 activity. However, it must be noted that inflammation is not the only factor in T1DM and other AAT responses should be investigated.

The *SERPINA1* gene is described to be a transcriptionally complex gene as it contains a number of different splicing events, including skipped exons, alternative donors, and alternative acceptors [66]. The *SERPINA1* gene promotor becomes hypermethylated in the later term of pregnancy, leading to elevated levels of *SERPINA1* expression [22]. AAT deficiency is suggested as a possible contributor to preterm premature rupture of membranes [67]. AAT protein expression levels are also controlled at the posttranscriptional level, by RNA structure influenced by noncoding gene regions [68]. It is also worth mentioning that there are several factors that could influence AAT activity rather than overall levels of gene expression and total plasma levels that could contribute to disease progression. Exposure to cigarette smoke can result in the oxidation of the methionine residues at sites 351 and 358 within the reactive center loop of the AAT protein, leading to diminished AAT anti-elastase function [69]. However, it was reported that AAT retains its anti-inflammatory properties even without its anti-elastase function [70]. Interestingly, serum levels of the AAT-oxidized low-density lipoprotein (LDL) complexes are high in smokers and decrease after smoking cessation due to weight gain [71] and could influence cardiovascular issues in emphysema patients [72]. AAT-LDL complexes also correlate with adiponectin levels in non-MetS subjects [73]. Circulating AAT undergoes glycation in hyperglycemic conditions and becomes inactivated [74]. The Z mutated form of AAT has increased fucosylation on N-glycans of Z-AAT that is associated with persistent inflammation [75]. AAT also circulates in high-density lipoprotein (HDL), as HDL was observed to be enriched with AAT and this may represent additional means for AAT to interact with many proteins [76]. AAT is secreted primarily from the liver and is distributed in plasma to many organs. However, where and when AAT incorporates into HDL is yet to be determined. AAT binds to unsaturated fatty acids, linoleic (C18:2) and oleic (C18:1), and the Z mutated form of AAT carries significant amounts of fatty acids [77]. This fatty acid-AAT complex induces expression of angiopoietin-like protein 4 (Angptl4) and fatty acid-binding protein 4 (FABP4), via a PPAR-dependent pathway [77], which could directly influence triacylglycerol homeostasis.

Finally, there is some mechanistic evidence suggesting that AAT could have a role in T2DM, with an imbalance between AAT and neutrophil elastase (NE) contributing to the development of obesity and insulin resistance in mice [78]. This was observed in NE knockout (*Ela2*^−/−^) mice and AAT transgenic mice, as they were resistant to high-fat diet-induced body weight gain, insulin resistance, inflammation, and fatty liver [78]. AAT regulated the AMP-activated protein kinase (AMPK) responses, fatty acid oxidation, and energy expenditure [78].

## 5. The Potential Use of AAT to Treat Diabetes

The only current safe treatment available for AAT deficiency-associated emphysema is intravenous AAT augmentation, which protects the lungs from disease progression [79]. On the basis of the potential benefits of AAT in ameliorating T1DM-associated inflammation and improving β-cell function, there are several clinical trials evaluating the potential for AAT therapy to treat T1DM. However, only a small number of these trials have published their findings and suggest certain populations of T1DM patients may benefit from AAT treatment (see Table 1). Small-sized studies in children and adults have demonstrated that AAT therapy is safe and well-tolerated in stage 3 T1DM [80] but a high dose may be required to see beneficial effects [81]. A recent phase II, double-blind, randomized, placebo-controlled clinical trial observed that AAT intervention reduced HbA1c levels in a subgroup of adolescents with recent-onset T1DM [82]. Children treated with AAT infusions have fewer IL-1β producing monocytes and dendritic cells [80].

Several animal and cell studies that explored AAT treatment reported AAT treatment reduces inflammation and retinal neurodegeneration via downregulation of NF-κB, iNOS, and TNF-α in the T1DM/Streptozotocin (STZ) mouse model of T1DM [83]. Equally, AAT can counter hyperglycemia-induced inflammation in retinal pigmented epithelial cells by controlling Akt responses [84]. In a rat insulinoma cell line, AAT increases insulin secretion in a glucose-dependent manner, while reducing TNF-α-induced apoptosis and cytokine production [31]. Recently, the application of CRISPR-Cas9 technology to correct AAT mutations in vivo was utilized to either correct an AAT mutation or enhance AAT production [85]. Equally, recombinant adeno-associated virus delivery of inducible human AAT significantly prevented T1DM development in NOD mice, and similar approaches could be utilized as future treatment approaches [86]. Therefore, AAT plays important roles in emphysema and T1DM, and possibly T2DM.

## 6. Conclusions

There is an emerging role of AAT in the onset and pathogenesis of diabetes mellitus especially T1DM and the use of AAT targeted therapies for the treatment of T1DM. These treatments could be effective on established T1DM but also might retard the progression of recent-onset T1DM. The role that AAT plays in T2DM pathogenesis is less clear and further research is needed to elucidate this association and possible therapeutic interventions.

## Figures and Tables

**Table 1 medicina-57-00397-t001:** Summary of clinical trials of Alpha-1 Antitrypsin in type 1 diabetes mellitus.

Study	Design	Population	Intervention	Main Outcomes
Gottlieb, P.A. 2014 [80]	Prospective, phase I, open-label interventional trial.	*N* = 12 (subjects with T1DM within ∼4 years from disease diagnosis.) Age: 24.6 ± 10.5 years (range, 12–39 years) Sex: 4 females and 8 males.	8 consecutive weekly infusions of 80 mg/kg of AAT (Aralast; Baxter Inc) were given.	No significant adverse effects were detected. Decreased total content of TLR4-induced cellular IL-1β. Improved β-cell function correlated with lower IL-1β production.
Rachmiel, M. 2016 [33]	Prospective, phase I/II open-label, interventional trial.	*n* = 24 (recently diagnosed subjects with T1DM Age: 12.9 ± 2.4 years (range, 9.8–17.6) Sex: 12 females and 12 males.	18 infusions of 40, 60, or 80 mg/kg/dose high-purity, liquid, ready to use AAT (Glassia^®^; Kamada Ltd.) over 28 weeks. 12 weeks: weekly. 12–20 weeks: once every 2 weeks, 20–28 weeks: once every 4 weeks	No serious adverse events were reported. Glycemic control parameters improved during the study in all groups, independent of dosage. Eight subjects (33.3%) that were considered possible responders had a shorter duration of T1DM) and a greater decrease in their HbA1c.
Weir, G.C. 2018 [66]	Prospective, phase I multicenter, open-label, dose-escalation study (RETAIN).	*n* = 16 (within 100 days of diagnosis of T1DM) Age: 8 adults aged 16 to 35 years and, 8 children aged 8 to 15 years)	12 infusions of AAT (Aralast NP; Baxter Inc): a low dose of 45 mg/kg weekly for 6 weeks, followed by a higher dose of 90 mg/kg for 6 weeks.	C-peptide secretion during a mixed meal remained relatively stable during the treatment period in adults and decreased in children. HbA1c and Insulin usage remained relatively stable during the treatment period on both groups but gradually increased afterward. AAT suppressed expression of genes involved in NF-κB activation and apoptosis pathways.
Lebenthal, Y. 2019 [82]	Phase II, Double-Blind, Randomized, Placebo-Controlled, Multicenter Study	*n* = 69 (recently diagnosed T1DM patients) Age: 13.1 ± 4.1 years Sex: 32 females and 37 males.	22 infusions of AAT (Glassia^®^; Kamada Ltd.) (60 or 120 mg/kg) or placebo.	AAT was tolerated well, with a similar safety profile between groups. C-peptide, glycated hemoglobin (HbA1c), and the total insulin dose (U/kg) were similar across groups. C-peptide AUC levels in the AAT-120 mg/kg adolescent group remained relatively stable in contrast to the decline observed in the placebo and AAT-60 mg/kg groups. The frequency of responders with at least 95% β-cell function reserve was 29% in the AAT-120 group and nil in the placebo.

AAT: alpha-1 antitrypsin; T1DM: type 1 diabetes mellitus; TLR-4: Toll-like receptor 4; IL-1β: interleukin-1 β, HbA1c: glycated hemoglobin A1c; NF-κB: nuclear factor-kappa B; AUC: area under the curve.

## Data Availability

Not applicable.

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
