# Peer review of "Therapeutic Potential of Alpha-1 Antitrypsin in Type 1 and Type 2 Diabetes Mellitus"

_medicina, 2021, doi:10.3390/medicina57040397_

Round 1

Reviewer 1 Report

This article reviews the application of AAT for the treatment of type 1 and type 2 diabetes. Although similar topic been reviewed in the past, this article included recent clinical studies. Therefore, it is of the general interests in the field. The authors may consider the following points:

  1. In line 37 on page 1, the fist paper showing AAT gene therapy prevent T1D {Gene Therapy (2004) 11, 181–186} should be mentioned and cited.
  2. In line 96 on page 3, b-cell should be β-cell. It should be consistent, not B-cell (line 102)
  3. In line 106 and 107 on page 5, the sentence is confusing and should be reframed.
  4. In Table 1, the brand name and the manufacture information of AAT used in each clinical study should be indicated.

Author Response

In line 37 on page 1, the fist paper showing AAT gene therapy prevent T1D {Gene Therapy (2004) 11, 181–186} should be mentioned and cited.

In line 96 on page 3, b-cell should be β-cell. It should be consistent, not B-cell (line 102)

In line 106 and 107 on page 5, the sentence is confusing and should be reframed.

In Table 1, the brand name and the manufacture information of AAT used in each clinical study should be indicated.

Response: We thank the reviewer for their constructive comments and have made the suggested changes.

Reviewer 2 Report

The review manuscript entitled “Therapeutic potential of alpha-1 antitrypsin in type 1 and type 2 diabetes mellitus” by Park and colleagues provide an overview on the potential role of alpha-1 antitrypsin as therapeutic agent in type 1 and type 2 diabetes mellitus. A summary of the current clinical trials of AAT in type 1 diabetes mellitus is also provided.

Strengths

The manuscript is in general well written.

Reference list is adequate.

Table 1 is detailed

Weakness

A molecular description of the AAT coding gene/gene expression regulation as well as main mutations linked to AAT deficiency should be included.

Major points

  1. The AAT coding gene is referred to as SERPINA1. SERPINA1 gene is highly polymorphic of well-known clinical interest as deficiencies of its AAT are associated to respiratory disorders. A brief paragraph describing the gene organization and its tissues specific regulation, might be helpful for the reader. For instance, different promoters within SERPINA1 gene provides gene regulation in a tissue specific manner (PMID: 26480348; PMID: 29109288; PMID: 33015055). Mutations of SERPINA1 are linked to AAT deficiency.
  2. AAT expression is influenced by the codominant expression of two SERPINA1 gene alleles: the normal M, the S which is related to AAT plasma levels of ~60% of normal, the severely deficient Z with AAT levels of ~15% of normal (PMID: 23776367; PMID; 14522813; PMID: 15978931). Since AAT deficiency is highly related to mutations of SERPINA1 coding gene, a description of the main mutations should be included.
  3. Serum AAT levels have been found to rise physiologically during pregnancy (PMID: 33015055), where inflammatory events might occur. ATT insufficiency has been hypothesized as possible contributor to preterm delivery (PMID: 21843112). These information/refs, which underline the role of AAT as anti-inflammatory factor, should be, at least briefly, included

Author Response

Strengths. The manuscript is in general well written. Reference list is adequate. Table 1 is detailed

Response: We thank the reviewer for their constructive and thoughtful comments.

The AAT coding gene is referred to as SERPINA1. SERPINA1 gene is highly polymorphic of well-known clinical interest as deficiencies of its AAT are associated to respiratory disorders. A brief paragraph describing the gene organization and its tissues specific regulation, might be helpful for the reader. For instance, different promoters within SERPINA1 gene provides gene regulation in a tissue specific manner (PMID: 26480348; PMID: 29109288; PMID: 33015055). Mutations of SERPINA1 are linked to AAT deficiency.

Response: As a researcher focused on respiratory disorders (including AAT deficiency) for the past 20 years, I agree with the reviewer on the importance of AAT deficiency. We have expanded on this here on lines 44-54 and 116-122.

AAT expression is influenced by the codominant expression of two SERPINA1 gene alleles: the normal M, the S which is related to AAT plasma levels of ~60% of normal, the severely deficient Z with AAT levels of ~15% of normal (PMID: 23776367; PMID; 14522813; PMID: 15978931). Since AAT deficiency is highly related to mutations of SERPINA1 coding gene, a description of the main mutations should be included.

Response: The reviewer is correct that many mutations of AAT exist leading to different plasma concentrations of AAT, with the most frequent and clinically significant alleles being the PI*M, PI*S, and PI*Z. We have now outlined this on lines 44-54.

Serum AAT levels have been found to rise physiologically during pregnancy (PMID: 33015055), where inflammatory events might occur. ATT insufficiency has been hypothesized as possible contributor to preterm delivery (PMID: 21843112). These information/refs, which underline the role of AAT as anti-inflammatory factor, should be, at least briefly, included 

Response: We have included these references and topics for consideration on lines 118-122.

This manuscript is a resubmission of an earlier submission. The following is a list of the peer review reports and author responses from that submission.